# Development of a peer-led, network mapping intervention to improve the health of individuals with severe mental illnesses: protocol for a pilot study

Jennifer Rose Deborah Collom,[1] Jonathan Davidson,[2] Daryl Sweet,[3] Steve Gillard,[4] Vanessa Pinfold,[3] Claire Henderson[5,6]

¹Institute of Psychiatry, Psychology and Neuroscience, King's College London, London, UK
²Health Service and Population Research Department, Institute of Psychiatry, Psychology and Neuroscience, London, UK
³McPin Foundation, London, UK
⁴Division of Population Health Sciences and Education, St George's University of London, London, UK
⁵King's College London, London, UK
⁶South London and Maudsley NHS Foundation Trust, London, UK

**Correspondence to**
Jennifer Rose Deborah Collom;
jennifer.collom@kcl.ac.uk

## ABSTRACT

**Introduction** Adults with severe mental illness (SMI) have reduced life expectancy and many have comorbid physical health conditions. Primary care providers are experiencing increased demands for care for people with SMI. Barriers to accessing physical healthcare have been identified which negatively affect quality of care. We propose that peer support workers (PSWs) could deliver an intervention to service users to promote their physical health by drawing on existing social support. The aim of this research was to pilot a novel PSW-led intervention, including personal well-being network mapping, to improve access to primary care for physical health needs.
**Methods and analysis** Twenty-four participants will be recruited from community-based mental health teams in two boroughs of London. Each participant will be offered a six-session intervention. Quantitative data will be collected before and after intervention (at 4-month follow-up). Qualitative interviews will be conducted with PSWs after completion of the intervention and with participants at a 4-month follow-up. Some intervention sessions will be observed by a member of the research team. This is a pilot study with a small sample aiming to assess acceptability and feasibility of an intervention. We aim to use the results to refine the existing theory of change and to optimise the intervention and its evaluation in a future randomised controlled trial. This study is strengthened by its potential clinical importance and origin in previous research where service users engaged with well-being network mapping.
**Ethics and dissemination** This study has been approved by the London-Chelsea Regional Ethics Committee (ref: 17/LO/0585). The findings will be disseminated to participants, the National Health Service trusts that we recruited from, primary care mental health leads, commissioners and in peer-reviewed journals and academic conferences.

## INTRODUCTION
### Background
People with severe mental illness (SMI) have greatly reduced life expectancy compared with the rest of the population, of 15–20 years depending on gender and diagnosis.[1] High rates of smoking, obesity and low levels

### Strengths and limitations of this study

► The research will employ a mixed methods design; the qualitative data collected from participants and peer support workers will aid the development of the intervention.
► The study will inform a larger scale research study into the intervention and, in turn, could inform the development of a new approach to physical healthcare for people with severe mental illness.
► Robust statistical analysis cannot be performed on the data due to the small sample size and no objective physical health measurements will be collected; this study does not aim to pilot collection of physical health data.
► The qualitative data collection employed by this study can be viewed as non-generalisable.

of exercise contribute to this,[2 3] as well as mental health-related deaths.[4] Engagement with primary care practitioners and facilities for health promotion (such as leisure facilities, smoking cessation groups, social activities) is increasingly important as many people with SMI have been discharged from community-based mental health services, and care pathways have been redesigned to accelerate discharge after both referral and re-referral from primary care. Research published in 2012 found that a high proportion of people with SMI are supported solely by a primary care physician (called a general practitioner (GP) in the UK) (31%), or with minimal specialist mental healthcare input.[5] This number is increasing as thresholds for admission to specialist mental health services are raised and more people discharged back to primary care.[6]

However, this group has generally poor access to primary care for physical health needs. Primary care within current service configurations is not best placed to absorb

increasing numbers of people with SMI discharged from secondary provision,[5] and there is wide variation in the extent to which primary care is proactive in ensuring access to care for physical health problems.[7]

Accommodations for those with SMI could be made within healthcare services, such as extended appointment times, to allow for discussion and treatment of both mental and physical health. Such accommodations are feasible within these settings and have been implemented for people with learning disabilities,[8] another group who experience poorer physical health and a lower life expectancy than the general population.[8]

Attitudes to people with SMI may create a barrier to responding to physical health complaints. Research has outlined how many UK GPs are less keen on supporting people with SMI than other medical conditions[9], lack specialist knowledge and skills[10], and have clear ideas that their role should be limited to physical health checks and providing medication.[11] [12] In the USA, Corrigan and colleagues found that primary care professionals working in the US Veteran's Health Administration, who endorsed stigmatising characteristics of a patient with schizophrenia described in a vignette, were more likely to believe the patient would not adhere to treatment; as a result, they were less likely to refer to a specialist or refill their prescription.[13] There is also evidence from the USA that family physicians are less likely to believe that patients with previous depression have serious medical conditions causing physical symptoms, leading to a greater reluctance to carry out investigations for such symptoms.[14] This may reflect the misattribution of physical symptoms to pre-existing mental illness,[15] a phenomenon known as 'diagnostic overshadowing'.[16] A qualitative interview study of emergency department nurses and doctors[17] revealed that this is a fairly well-recognised problem that can lead to adverse consequences varying from delay in treatment to death. This study also found that some participants avoid people who are experiencing symptoms of mental illness due to fear of violence, which may also adversely affect quality of care.[17] Fear of patients with substance abuse problems has also been expressed by district nurses who work in primary care, with the consequent risk of suboptimal care.[18]

## Peer support

Supported access and empowerment are thus needed for improved physical healthcare in this group. These functions may be carried out by professionals in specialist mental healthcare; however, caseload size may not permit the time needed to focus on physical health and those whose mental health is fairly stable are likely to be discharged to primary care.[19] Evidence from a number of trials reviewed recently suggests that some practitioner roles can be performed by trained and supervised peer support workers (PSWs), leading to similar outcomes as when they are undertaken by mental health staff.[20] PSWs are people with their own experience of mental health problems who use this experience to help others. They

are becoming increasingly frequently employed within health services.[21] There are also potential longer term advantages of employing peer workers in this role; it creates a rung on the career ladder for the peer, which is also visible to the person in receipt of their support, encouraging recipients to, in turn, become a peer supporter or take on other similar roles.[22]

Recent systematic reviews have evidenced the benefit of peer-led interventions on physical health outcomes for people with SMI. Authors found studies measuring service use, self-related health and quality of life among other outcomes.[23] [24] They concluded that the most beneficial interventions focused on self-management of health behaviours and that peer-led interventions had the most significant impact on hope and quality of life; these reviews support the outcomes measured in the current study and the need for well-developed peer-led interventions.

This project draws on peer-led[25] [26] and peer co-facilitated[27] interventions in the USA to improve self-management of physical health and increase uptake of primary care services for physical health conditions, and on UK research exploring social networks and their potential resources to support people with SMI. It is important to investigate peer support interventions for physical health in the UK, which has a different health system and different peer support services from the USA. In the UK, the first well-being network mapping was developed by the McPin Foundation[28] following their study of social networks and how they relate to personal well-being.[29] Several other studies[30–32] have drawn on this and other work on social networks to seek to address social isolation, by encouraging network development or enhancement using workers to empower and signpost individuals to opportunities locally to build connections to people, places and activities. Support networks are made up of social contacts, and recovery is linked to meaningful activities, identity and sense of purpose in life, empowerment, hope and connectedness. Further, community places and spaces constitute the environments in which people look after their health and well-being and build social networks. Well-being network mapping is asset based, strengths focused and through structured conversations may facilitate engagement and planning to address identified health and social care needs. The use of this tool will allow PSWs to assess the client's relationship with primary care, in the context of other health-related behaviours and relationships, which can promote or adversely affect health. The PSWs will work with their client to use the connections established using the map with the aim of improving well-being in a sustained way, and a focus on access for physical health needs.

The project adopts a theory of change (ToC) approach.[33] [34] This is a method for developing a framework for delivering change and explaining the mechanisms for doing so. It covers assumptions, inputs and mechanisms, and suggests appropriate outcomes based on underlying processes in order to better understand what is involved in achieving sustained long-term

change. Previous work on well-being network mapping has not employed this approach. It is hoped that the ToC will improve current understanding of how people move from identifying resources and barriers in their networks to making positive changes to their physical health and access to healthcare.

## Aims

The aims of the project are as follows:

1. To develop a localised model for a peer well-being network mapping intervention in two boroughs.
2. To produce a ToC, mapping out assumptions, inputs, mechanisms and outcomes.
3. To deliver the intervention in two boroughs and assess impact on people with SMI and the PSWs.
4. To inform the development of a future feasibility trial to assess feasibility of a pragmatic trial of the effectiveness of the intervention with respect to access to primary care for physical health needs.

This paper describes work done for aims 1 and 2 and the protocol for aims 3 and 4.

## METHODS AND ANALYSIS
### Design

This study will pilot an intervention to improve physical healthcare using well-being network mapping delivered by PSWs. The intervention addresses barriers to accessing physical healthcare and employs behavioural activation through goal setting for people with SMI. All participants will receive the intervention. The research will be conducted in three stages, the first of which is complete. Stage 1 comprises development of the intervention through key informant interviews and consulting a working group and producing a preliminary ToC. Stage 2 comprises observing the training of PSWs and use of well-being network mapping, pilot mapping interviews and piloting the intervention (n=24). Stage 3 will include a stakeholder workshop and finalisation of the ToC model.

### Settings

The intervention will be conducted across two London boroughs (local government areas). Participants will be recruited from community-based mental health teams providing care for people with SMI. In one borough, the four teams we will recruit from specifically provide services for people with psychosis. In the other, the three teams work with people with psychosis and other conditions such as depression and anxiety.

### Stage 1: development of the intervention

Working group: A working group was set up of five people including members of the research team; a representative from each of the other network mapping projects, that is, a community navigator from Hounslow and the principal investigator (PI) of the Community Navigator Study; and a GP working in one of the boroughs we will recruit from.

The working group met during the development phase of the study.

Key informant interviews: In each site key informant interviews were held with primary care mental health leads and peer support service managers to identify how the intervention would link with other provision and any clinical governance arrangements to be put in place. The PI liaised with mental health services through attendance at team meetings. The aim of this was to introduce and generate a sense of ownership over the idea of the intervention, consult on the arrangements identified through the key informant interviews, discuss how to identify potential participants and agree the arrangements with any amendments needed.

Prepilot of the mapping process: A manual was drafted by the research team and PSWs who helped to refine and streamline the process. It was then tested by PSWs on each other and with the researcher. The research team observed and documented this process; data were field notes, interview transcripts and workshop outputs.

During the development of the intervention, two goals for the service user to work towards were added. The first goal was to visit a primary care service for a physical health appointment; the second was to create a behaviour change with a view to improving physical health. The aim was for the peer to encourage service users to use the map when working towards these goals, as well as supporting them to attend an appointment by offering to accompany them.

### Theory of change

The development of the intervention will be informed by a ToC model (figure 1). This will help identify the desired impacts and outcomes and the mechanism by which these are achieved. To refine the ToC for further testing, we will use a mixed methods approach, collating quantitative outcome measures (before and after intervention), and qualitative interviews and focus groups with both participants and PSWs. We will also conduct weekly supervision with PSWs and observe some of the intervention sessions.

### Stage 2: piloting
#### Recruitment

A member of the research team will approach clinical staff at community-based mental health team centres. They will be informed about the research, the inclusion/exclusion criteria and given an information script. This will provide clinical staff with details of the study to relay to possible participants. The staff will be asked to consider their entire caseload, which usually consists of up to 30 people, and to contact eligible service users as agreed with the Research Ethics Committee. When clinical staff receive agreement from a service user, a meeting will be organised; during which, a researcher will provide a patient information sheet and answer any questions. Participants will be offered a period of time, usually at least 24 hours, to consider their involvement in the study according to Integrated Research Application

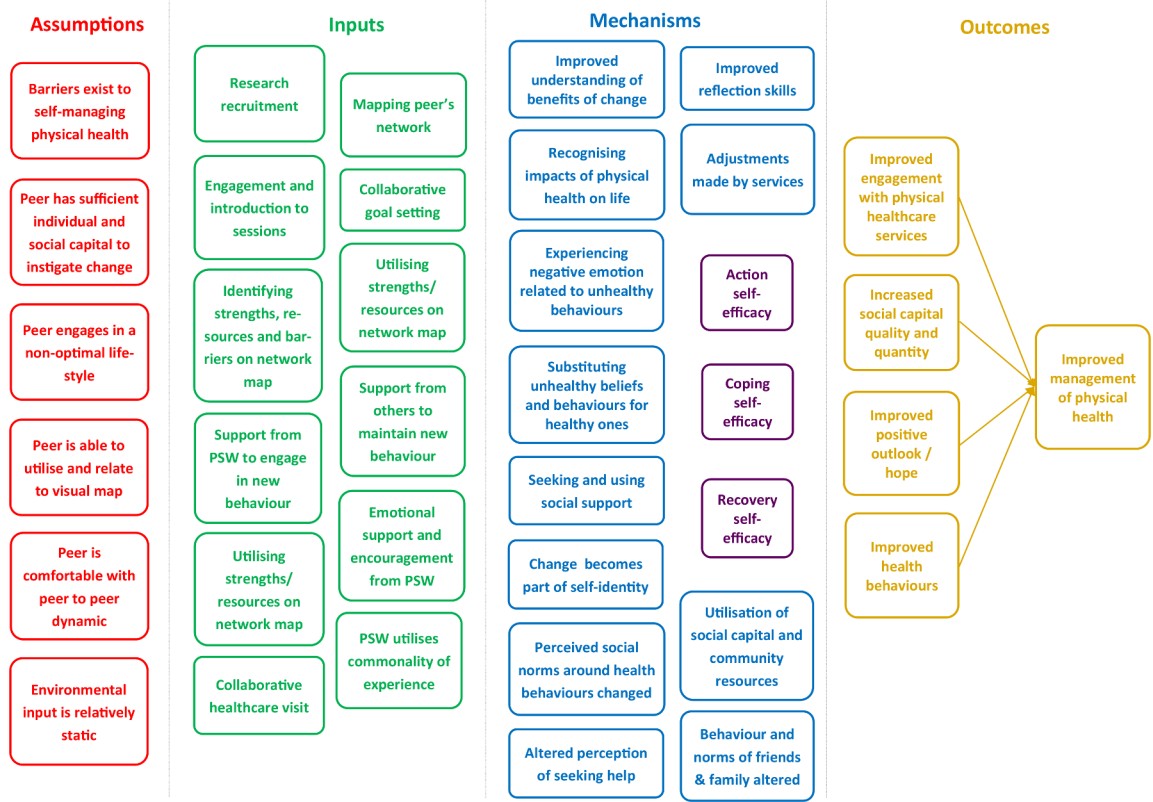

**Figure 1** Preliminary theory of change model. PSW, peer support worker.

System guidance[35] before providing consent to participate in the study.

## Sample size

The target sample size of 24 is based on the pragmatics of testing the feasibility of delivering this intervention to inform a larger scale study; therefore, this sample size is appropriate.[36]

### Inclusion and exclusion criteria

The following inclusion criteria will be used: (1) approaching discharge from community-based mental health services, as this group is soon to cease contact with mental health professionals who encourage healthy behaviours and help seeking for physical health; (2) at risk of deterioration in physical health due to poor access to primary care; (3) demonstrated failure to organise/attend primary care appointments for comorbid physical health difficulties; (4) aged 18 or older; (5) clinical diagnosis of severe mental illness (a schizophrenia spectrum disorder or bipolar I).

People will not be eligible to take part in the study if they (1) lack capacity to provide informed consent; this will be assessed during the consent process as the ability to retain, weigh up and make decisions based on the information provided; (2) pose a high risk of harm to others, as judged by their clinical team; (3) are unable to communicate in English; (4) are currently an inpatient in a psychiatric hospital; and (5) non-attendance of discharge planning meeting with subsequent discharge.

## Patient and public involvement

PSWs, who have lived experience of mental health problems, were involved in the development of the intervention through their provision of feedback on the draft manual, and will deliver and feed back their views on the intervention. We also brought together a group of PSWs and others involved in projects with the McPin Foundation that include a mapping component as advisors. They helped refine the mapping process, contributed to the planned training programme and inputted into the draft ToC.

## The intervention

Peer support workers: PSWs (n=6) were recruited from within existing peer support services in two London boroughs (three per borough). The PSWs were paid to take on the peer well-being role. We aimed that each PSW will work with up to four clients. The training took place across 2 days. The first day focused on the intervention, including the PSWs carrying out the mapping intervention with one another, and development of the manual. The second day was a refresher on the key aspects of peer support. Training was delivered by a network mapping researcher (DS), a researcher with expertise on peer support (SG) and a researcher who lead on intervention development with input from the whole study team (JD). PSWs will be supported with use of the well-being mapping tool through regular supervision with the research team and their line manager.

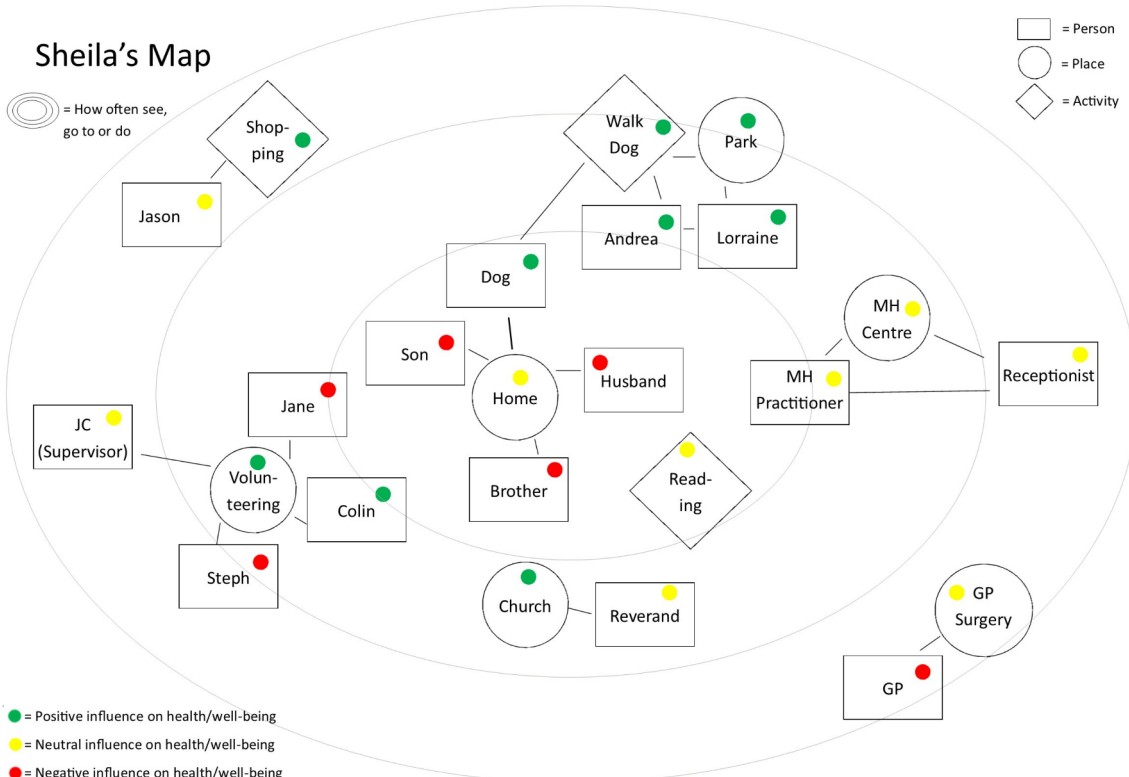

**Figure 2** Example of a network map. GP, general practitioner.

Over six sessions, the intervention aims to cover the following topics: engagement of the client with the peer and the intervention; mapping of the client's network (see figure 2 for an example map), including people and services which contribute positively, negatively or both to the client's well-being; goal setting for physical health; and planning a visit to a primary care service, for example, their GP's surgery, dentist or sexual health clinic, to which the client may or may not be accompanied by the peer; monitoring of work towards the goals and review of the primary care visit; and an ending, to include a review and summary of the client's achievements, updating the well-being map and encouragement to continue to use it. The visit to a primary care service will be prepared for collaboratively using resources that aim to prevent diagnostic overshadowing. For example, topics to be discussed in the appointment will be pre-agreed between the PSW and service user.

Some people with SMI may not have robust or wide social networks. In these cases, the PSWs will make suggestions for ways to expand the person's social network. The observations, notes from supervision with PSWs and qualitative feedback will identify whether additions to the intervention are needed to try to build social networks.

The network mapping process contributes to reflexivity, allowing participants to become more aware of the resources they have in networks to support physical health and make positive behavioural change. Physical health can be activated indirectly through person-centred goals. A person might not be motivated to address physical health concerns, but they might be motivated to address an area of the well-being network map that matters to them. The peer can use these personal goals to indirectly improve the individual's physical health goals. For example, if they wish to join a book club to meet new people, the peer could suggest they walk/cycle to this club.

Participants can arrange to meet with their PSW in a location that suits them both. This may include private rooms at their community-based mental health service, a library or a café. The six sessions will, in most cases, take place across 6 weeks; however, this time period can be elongated to reflect the needs of the service user and to allow a primary care visit. Each session will last up to 2 hours. The sessions are guided by the manual; however, the map and the goals created will be personalised to the individual.

### Baseline and follow-up procedures

Baseline data collection will take place following consent and before the initial meeting with the PSW. The participants will meet with a PSW and begin the intervention. After finishing the intervention, the participants will be asked to complete outcome measures at 4 months from baseline, before completing a semistructured qualitative interview. PSWs will also take part in a semistructured qualitative interview and a focus group. The qualitative interviews will document the participants' view of their physical health and its management, experiences of the intervention, ways in which it worked, identify mechanisms through which the mapping helped lead people to take action on their physical health, and factors

that limited its helpfulness and how it could be improved. A focus group with all PSWs across both sites will seek feedback on use of the mapping tool; discuss the actions planned versus those taken, and reasons for deviation; problems encountered and methods for solving them; and the outcomes achieved.

The participants will be compensated £10 for each of their baseline and follow-up assessments and interviews. A further £15 will be given for focus group participation.

## Outcomes

The peer support change model developed by Gillard et al[37] was used to identify outcomes, and hence, the following outcomes will be measured before intervention and at a 4-month follow-up after the intervention has been completed.

Service use and demographic information (age, sex, marital status, ethnicity, nationality, language, education, employment and housing status) will be measured using an adapted version of the Client Service Receipt Inventory.[38] Health and social care, informal care and use of leisure activities were measured. Primary care and secondary care (for physical healthcare) will be used as intermediate outcomes on the pathway to improved physical health; likewise, leisure activities with potential benefit for physical health such as those involving exercise.

Social capital will be measured using the Resource Generator UK[39] (RG-UK). The RG-UK asks the participants if they have or could obtain access to each of 27 skills or resources within their social network within 1 week. It then asks the nature of the social tie through which they could access each skill or resource. The instrument has four subscales: domestic resources, personal skills, expert advice and problem solving resources. This measure assesses service use and healthy behaviours to capture improvements to management of physical health. It reflects reciprocal relationships, so change may be due to either the participant, members of their network or both. It has good reliability and validity, and has been used in samples of people with mental health problems, for example, producing valid results. The RG-UK does not fully measure access to resources related to health, the primary interest of this study. Consequently, an additional subscale for the RG-UK was used, developed by Pinfold et al,[40] which captures health and well-being-related resources.

Social network quality and quantity will be measured using the Lubben Social Network Scale, a six-item self-report measure assessing quantity and quality of contact with family and friends.[41] Three questions to capture acquaintances were added to this measure.

Hope will be measured using the Herth Hope Index,[42] a 12-item self-report measure designed to measure levels of hope in adults in clinical settings, increasingly used in mental health studies. This measure has been found to hold divergent and construct validity.[42]

Mental health self-efficacy will be measured using the Mental Health Confidence Scale,[43] an 11-item self-report measure based on theories of self-efficacy and self-help. The scale was shown to have good internal consistency and clinical use.[44]

Mental well-being will be measured using the Short Warwick-Edinburgh Mental Well-being Scale,[45] a seven-item measure designed to measure mental well-being and found to have good reliability and validity for people with SMI.[46] The positively worded questions focus on assessing mental well-being as opposed to mental health difficulties.

Physical health will be measured using the EuroQol Five Dimensions Questionnaire,[47] a two-part questionnaire assessing the objective and subjective health status of the participants. The measure can be used to compute a Quality Adjusted Life Year,[48] an outcome frequently used in cost analyses and has been found to be a valid measure among people with schizophrenia.[49]

Some outcomes will not be measured, such as emotions relating to unhealthy behaviours, health beliefs, self-identify, social norms or behaviours of friends and family. These are intermediate outcomes and we have prioritised outcomes further along the ToC model to reduce respondent burden.

## Process evaluation

With the permission of both PSW and participant, the project researcher will observe PSWs delivering some sessions of the intervention to participants in vivo. The aim of this observation was to evaluate the extent of the use of the mapping tool; whether and, if so, how it is used to inform goal setting and planning; and whether its use creates any difficulties in goal setting and planning. We will record information provided during supervision sessions between the research team and the PSWs.

## Proposed data analysis

Quantitative data: Feasibility of the intervention will be assessed by rates of recruitment of peers and of clients at each site; implementation of the intervention—that is, what proportions of participants had each of zero to six sessions with a peer and what proportion visited their GP or other primary care service for a physical health issue. Feasibility of the evaluation will be assessed by completion rates of interviews and measures of peers and clients; rates of attendance at focus groups. Floor to ceiling effects of measures will be identified, as well as the level of outcome measure completion.

Qualitative data: Data from the first two participants at each site will be rapidly analysed by the study researchers with the main aim of identifying any feasible modifications needed to the intervention or study design. We will use a focused thematic analysis approach[50] to identify contextual barriers and facilitating factors to the successful implementation of the intervention with respect to the desired outcomes, the appropriateness of the selected outcomes (above) and mechanisms of action of the intervention. Multiple coding will be conducted on five transcripts to allow researchers to identify and discuss any alternative interpretations. The study researcher will

input all coding using NVivo, with crosschecks by another team member. We will hold analysis meetings to review transcripts and develop coding frames. Further synthesis meetings will be required once coding is complete.

### Stage 3: collated feedback

Stakeholders from both boroughs will be brought together to discuss the results. The outputs following the workshop will include (1) agreed documentation for use including guides for both PSWs and clients to support the model; (2) select and refine outcomes for a randomised controlled trial; (3) manuals to document the well-being mapping process—one for clients and one for well-being partners.

### Ethics and dissemination

Ethics approval for this study has been obtained from the Heath and Research Authority and the London-Chelsea Regional Ethics Committee (ref: 17/LO/0585).

We plan to disseminate the results to the participants who received the intervention and the PSWs who delivered the intervention. We will also share the results with the National Health Service trusts we recruited from, the peer support services from which the PSWs were recruited, primary care mental health leads and commissioners. We plan to publish the findings in peer-reviewed journals and share our findings at academic conferences.

**Acknowledgements** The authors would like to thank Bryn Lloyd-Evans, Amanda Perry, Sally Gomme (Hounslow), Nancy Kuchemann (GP and Southwark CCG clinical lead) and the Southwark and Sutton peer support services.

**Contributors** JRDC was the research assistant responsible for recruitment of participants to the study, data collection and peer supervision. JD was the research assistant responsible for helping develop the intervention, train the peer support workers and recruited participants to the study. CH was the chief investigator. DS and VP helped to develop the intervention. DS helped to train the peers. SG was the site lead and helped to train peers. JRDC drafted the paper and all authors provided edits and comments for its revision.

**Funding** This research was supported by the Maudsley Charity and the National Institute for Health Research (NIHR) Collaboration for Leadership in Applied Health Research and Care South London (NIHR CLAHR South London) at King's College Hospital NHS Foundation Trust. The views expressed in this article are of those of the authors and not necessarily those of the NHS, the NIHR or the Department of Health and Social Care.

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
