## [Reviewer comments · BMJ Open]

ARTICLE DETAILS

TITLE (PROVISIONAL)	Development of a peer-led, network-mapping intervention to improve the health of individuals with severe mental illnesses: protocol for a pilot study
AUTHORS	Collom, Jennifer; Davidson, Jonathan; Sweet, Daryl; Gillard, Steve; Pinfold, Vanessa; Henderson, Claire

VERSION 1 - REVIEW

REVIEWER	Candelaria Mahlke University Medical Centre Hamburg-Eppendorf (UKE), Germany
REVIEW RETURNED	27-Aug-2018

GENERAL COMMENTS	This is a very thoroughly developed and planned intervention and study. The topic is timely and interesting and adds to the growing literature on PSW roles and options for empowerment and the connection to health care. I'm looking forward to reading the results. The manuscript is very well written. There were only some minor questions I had reading it and a few suggestions I would like to make. Introduction: May I suggest some literature you could refer to, or might be of your interest: Cabassa, L.J.; Camacho, D.; Vélez-Grau, C.M.; et al. (2017): Peer-based health interventions for people with serious mental illness: a systematic literature review. J Psych Res, 84, S. 80–89. Bartels, S.J.; Aschbrenner, K.A.; Rolin, S.A.; Hendrick, D.C.; Naslund, J.A.; Faber, M.J. (2013): Activating older adults with serious mental illness for collaborative primary care visits. Psychiatr. Rehabilitation J, 36, S. 278-288. Bellamy, C.; Schmutte, T.; Davidson, L. (2017): An update on the growing evidence base for peer support. Mental Health and Social Inclusion, 21(3), S. 161-167. Wang et al. (2017) Social isolation in mental health: a conceptual and methodological review Soc Psychiatry Psychiatr Epidemiol. Anderson K, Laxhman N, Priebe S (2015) Can mental health interventions change social networks? A systematic review. BMC Psychiatry 15:297. Leigh-Hunt N, Bagguley D, Bash K, et al: An overview of systematic reviews on the public health consequences of social isolation and loneliness. Public Health 152:157-171, 2017.
---

Holt-Lunstad J, Smith TB, Baker M, Harris T, Stephenson D (2015) Loneliness and social isolation as risk factors for mortality: a meta-analytic review. *Perspect Psychol Sci* 10(2):227–237.
doi:10.1177/1745691614568352

Sokol, R.; Fisher, E. (2016): Peer Support for the Hardly Reached: A Systematic Review. *American Journal of Public Health*, Epub <https://doi.org/10.2105/AJPH.2016.303180a>

Recruitment:

L 31: who exactly asks the staff than – everybody, that falls into the inclusion criteria? How do you make sure there is no selection on the staff's behalf who to involve?

L 39: participate?

Sample size:

Can you give some information on how you determined the sample size? – Ahh I found it further down...I would move it up here instead of having two passages on sample size

Inclusion criteria: How do you define and assess the criteria 2 and 3? Is it any health condition and how do you identify the condition is a result of a poor access to primary care?

Exclusion criteria: Criteria 1 and 2: how do you assess this? I'm not sure if this is necessary as you wouldn't be discharged if you are at high risk of harming others.

Intervention:

L 16/17: I'm not sure if get the sentence right: there are n=6 PSW's recruited, but 3 and 4 respectively in the two boroughs– what happened to the seventh?

L 21: Can you give some more information on the training: by whom exactly? Have the trainers lived experience on their own? How long was the training, core elements etc.

Overall, can you give some more information on the 6 sessions: how flexible is the setting? Is there a duration/time limit for the sessions and how are they distributed over the six months – is it up to the participant? Where do they usually meet?

Development of the intervention:

Very nice, great work! You could brag a little bit with this in the introduction and abstract already I think.

Stages:

If I get it right, you already fulfilled the first stage? The intervention development and theory of change model is already a result than? The study protocol is for stage two and three? This seems a bit mixed up here – please clarify.

Baseline and follow up procedures:

Why is the follow up at 4 months if the interventions duration is 6 months? Or is the duration of all interventions spread over 6 months – please clarify in the Intervention section the planned duration of the intervention for one participant than.

Outcomes:

L 43: I'm not a native speaker obviously; but is there some punctuation or "it" missing here?

	L 46: added by whom? Maybe one item as example. Is something known on the validity of the Herth Hope Index, Mental Health Confidence Scale, Short Warwick-Edinburgh Mental Well-being Scale? On the EuroQol there is, could be useful to add this information. Process evaluation Further up in "Stages" it states that the training is observed – is this the observation of the intervention mentioned here? Data analyses: Please give some more information on the quantitative data analyses - there will be some descriptive analyses of the questionnaires? Ethics and... L 23:"PWS`s" Reference list: Reference 21 - something went wrong here, I think it is: Moran GS, Russinova Z, Gidugu V, Yim JY, Sprague C. Benefits and mechanisms of recovery among peer providers with psychiatric illnesses. Qual Health Res. 2012;22: 304–19. doi: 10.1177/1049732311420578 Check Reference 25, 26...hmmm maybe check the whole list again, it seems to me that there is some inconsistency.
--	--

REVIEWER	Erin Kelly UCLA, USA
REVIEW RETURNED	28-Aug-2018

GENERAL COMMENTS	Thank you for the opportunity to review the paper entitled "Managing transition from secondary mental to primary health care using peer-delivered well-being network mapping: protocol for a pilot study" The overall approach to this intervention seems reasonable and could have benefits for this high need population. However, the grammar and order of the article could use careful revision so that it's easier for readers to understand the proposed intervention. There are also key elements of the intervention that are not described that are important to include. For example, is the intervention delivered one on one or is it group-based? My first suggestion would be to adjust the title of the paper to something less convoluted. Abstract: Line 5- Severe mental illness is not typically capitalized in the text. Line 13- Why is Personal capitalized? Lines 15-17 – Is oddly stated: We aim to deliver the intervention in two boroughs; create a theory of change; and to inform the design of a future larger scale trial of the intervention. The first part does not seem well-related to the subsequent phrases and does not seem important to the introduction (should keep to methods). Do mean to use the theory of change to inform the design of a future study? What is going to be informed?
--

Line 23- Suggest rephrasing to "Twenty-four participants will be recruited by community mental health teams in two boroughs of London."

Line 33- another odd phrasing: This study is strengthened by its potential clinical value and origin in previous research where service users engaged with well-being network mapping.

Introduction

Overall, the introduction hits many relevant points but could be written more concisely and organized in a more logical manner.

Specific suggestions are below but are not exhaustive:

Reference#1 – there are numerous meta-analyses on this topic that are more suitable to cite here.

Page 5. Line 12 – there are more than just lifestyle factors that should be referenced as contributing to their early mortality

Page 5. Line 18 - Secondary care mental health services are not a universal term – are these community based mental health services? Is it an alternative term for specialty care? Please define for an international audience.

The first two paragraphs could weave together the issues facing those with SMI and their health and healthcare more comprehensively.

Page 5. Line 37 – the intended goal of the paragraph should be at the start of the paragraph. For example, Accommodations for those SMI could be made within health care services, such as extended appointment times. Explain why it would be beneficial to those group. These accommodations are feasible within these settings as they already exist for another special needs group, those with learning disabilities.

Line 46- There is a lot of extraneous information about the issue of mental health stigma in this paragraph that could be written more succinctly and combined with the subsequent paragraph.

Page 6- I would spend a little more time explaining and defining peers and what research has supported their use as everyone may not be as familiar with this literature.

Page 6. Line 46 - The description of the care navigator intervention is not well introduced. What is the rationale for the social network mapping for directing the intervention that you will describe? The whole paragraph could be reorganized as it alternates between listing the projects that you are using as inspiration to partially describing what they did.

Page 7 line 7 – You still have not established what personal well-being network mapping is and how you think it will be useful to your intervention. The authors should establish the definition and the importance of social networks to this intervention, then describe your intervention approach. Many of the elements are there but not in a logical order that builds your case for the reader.

Methods

The intervention sounds like it could be an interesting and effective approach, but the way that it is presented could be much improved with careful editing and reorganization. The overall size and scope of the pilot seem appropriate to the stated aims.

The figure of the theory of change could be improved by creating subcategories within the domains so that it's easier to follow. The authors did so with self-efficacy in the mechanisms section but systematically doing with other subcategories could be helpful.

Inclusion/exclusion criteria- why do participants need to be approaching discharge from mental health services?

The intervention:

	Line 16- you state that you have 6 peers across two clinics and then state “three and four PSWs respectively” – this does not make sense. Can you provide an example figure of a network map? I am curious about the choice of a 4-month intervention period. How did the authors decide on this length of time? It is likely to be challenging to detect significant changes in the health of your participants over such a short period of time. Measures – the proposed measures do not appear to capture all the stated mechanisms as written. What scale will detect negative emotions related to unhealthy behaviors? The Short Warwick-Edinburgh Mental Well-being Scale? I see services use but not health beliefs captured. Which measure will capture those? How will you capture change as part of self-identity? Social norms around health behaviors? Behaviors of friends and family? These elements may all be there but the inconsistency in terms makes it challenging to evaluate? Are changed behaviors of friends and family detected through the Social Capital measure? What is captured in the final outcome of "improved management of physical health"? Additional issues: It is unclear from your study design if this intervention is manualized, if it's conducted one on one or if it's group-based. Will the sessions all occur at the mental health clinic? Are they in vivo? How long are the sessions supposed to last? Are the sessions individualized or prescribed? How will you capture the number of completed sessions? The authors might want to include a measure of their relationship to the peer. Will this intervention address the diagnostic overshadowing issue raised at the beginning? Many individuals with SMI can have small or non-existent social networks - how will you use social network mapping to help those without robust social networks? Proposed data analyses: Your quantitative analytic plan does not include an evaluation of the measures to detect differences before and after the intervention. Rather, it appears that the only goal is to see if it's feasible for participants to complete these measures. As previous pilots have done, you should be able to test for within-person changes on these measures and used to estimate effect sizes, which will help you determine the sample size you need for a fully powered RCT.
--	--

REVIEWER	Alexandros Maragakis Eastern Michigan University
REVIEW RETURNED	10-Sep-2018

GENERAL COMMENTS	Overall I appreciate the work the authors are doing and I commend them for engaging in this work. However, there are several key issues that come up when reviewing their protocol: First, it is unclear to me who the actual target population is for the pilot intervention. Serious mental illness is a very large umbrella
--

	term, and I would advocate that for this pilot to be successful, the authors would need to narrow the scope of the population or assess for baseline functioning. For example, someone with schizophrenia in comparison to someone with bipolar or major depression may benefit this type of intervention in different ways based on their level of functioning. Given the different levels of general functioning within the SMI community, and the low N in this pilot, either diagnoses, or preferably, baseline functioning would need to be captured. Second, the current manuscript is unclear on how this is different than other peer-support interventions. Is the major difference that this pilot is occurring in the U.K. rather than the U.S.? Given the data on utility of peer interventions across multiple healthcare settings, what is to be gained from this pilot? Third, the authors are list various outcome measures, but only discuss feasibility in their data analysis. What is the plan for the rest of the outcome measures?
--	---

VERSION 1 – AUTHOR RESPONSE

Reviewer 1

Reviewer: suggested literature.

Response: we have added three of the recommended studies. We did not include the research focused on loneliness as this is not the focus of the intervention, though we hope loneliness will be improved by the mapping process.

Reviewer: how do you prevent selection bias from the staff.

Response: we have described the recruitment process in more detail to explain how staff are required to consider their entire case load (usually 30 people) as per Research Ethics Committee rules.

Reviewer: sample size – duplicated information about sample size.

Response: we have deleted the second section.

Reviewer: inclusion/exclusion criteria – how do we define and assess the criteria and how is the condition a result of poor access to primary care.

Response: we have added more information about inclusion criteria. The physical health condition is not deemed a result of poor access to primary care. Apologies, as this may not have been clearly stated. The condition is however likely to deteriorate with poor access to primary care.

Reviewer: the intervention – number of peer support worker is unclear.

Response: we had one drop out therefore six worked on the study. We have amended this section.

Reviewer: more information on training.

Response: we have added more detail about what the training covered and who it was delivered by. We do not know the lived experience of those who delivered the intervention so cannot comment. We had also added more detail about where the intervention occurred and time periods. We have added

a section on patient and public involvement which describes the involvement of people with lived experience of mental health problems in developing the training manual as well as a focus group which informed training and the theory of change.

Reviewer: stages – clarify the stages.

Response: we have amended the 'aims and objectives' to make it clearer.

Reviewer: baseline and follow up – timeline unclear.

Response: the intervention duration was typically 6 weeks, though sometimes longer due to participant preference; we have made this clearer.

Reviewer: Outcomes – validity of measures.

Response: we have added references to demonstrate the validity of the EuroQoL Five Dimensions Questionnaire, Short Warwick-Edinburgh Mental Wellbeing Scale, Mental health Confidence Scale and the Herth Hope Index.

Reviewer: Process evaluation – observation of training.

Response: we observed the peer support workers practicing delivery of the intervention whilst in training and then again when delivering it to participants in vivo. We have stated this more clearly in the paper.

Reviewer: more information on quantitative data analysis.

Response: we have explained that we will identify whether there are any floor or ceiling effects for any of the measures and assess their level of completion.

Reviewer: reference list – some errors.

Response: we have reviewed and amended the reference list.

Reviewer: key elements of the intervention not described e.g. group based or one-to-one.

Response: the intervention was delivered on a one-to-one basis; we have amended the description to clarify this.

Reviewer 2

Reviewer: Abstract – some odd wording and unclear aims.

Response: we have worded the abstract more appropriately and clarified that we aim to deliver the intervention, evaluate the process and to use our results to refine the theory of change and optimise the intervention and its evaluation in a further RCT.

Reviewer: introduction – more recent reviews of the literature.

Response: we have added references to recent systematic reviews of peer led interventions for physical health improvements in people with mental illness.

Reviewer: introduction – 'secondary mental health service' is not a universal term.

Response: we have amended to 'community based mental health services.'

Reviewer: introduction – the first two paragraphs could combine to cover issues faced by those with SMI and their health care.

Response: we have done this.

Reviewer: Page 5. Line 37 – state the aim of the paragraph at the beginning.

Response: this paragraph has been rephrased as suggested.

Reviewer: Line 46 – extraneous information about stigma.

Response: this has been rewritten more succinctly.

Reviewer: Page 6 – more information about peer support workers.

Response: we have described their role more clearly and cited a useful paper.

Reviewer: page 6. Line 46 – rationale for network mapping for directing the intervention.

Reviewer: this has been described more clearly.

Response: Page 7. Line 7 – description of network mapping remains unclear.

Reviewer: this description has been amended to add clarity.

Response: methods – theory of change figure could be improved.

Reviewer: the model in the paper is our preliminary model which will be revised upon analysis of our results.

Response: inclusion/exclusion criteria – why do participants need to be approaching discharge?

Response: we wished to focus on those approaching discharge as they will soon lose contact from their mental health service who may intervene directly in physical health problems or encourage help seeking. Primary care will become their central hub following discharge, and on approaching discharge they are most likely in a stable state that is amenable to intervention. We have edited this section to clarify this.

Reviewer: Line 16 – number of peer workers across sites.

Response: we have amended this section to clarify.

Reviewer: example figure of map.

Response: we have added an example of a social network map. This example was provided to participants when given the information sheet to inform them about the intervention.

Reviewer: measures – they do not capture all of the mechanisms.

Response: we have acknowledged that we do not measure for all of the mechanisms outlined in the theory of change model. Some of which are intermediate outcomes; we prioritised those further along the causal chain to reduce response burden.

Reviewer: measures – are changed behaviours of friends and family captured in the Social Capital measure?

Response: the measure reflects reciprocal relationships so a change may be due to either the participant or the members of their network or both. This measure also captured participant

perception, therefore they may perceive certain resources are now more available from existing or new contacts.

Reviewer: what is captured in the final outcome of “improved management of physical health?”

Response: this comprises service use and healthy lifestyle behaviours.

Reviewer: it is unclear if the study is manualised, one-to-one or group based.

Response: we have added more detail to make this clearer.

Reviewer: measure of relationship to peer.

Response: this is assessed qualitatively.

Reviewer: will the intervention address diagnostic overshadowing.

Response: yes, we do aim to do this by providing resources for the peer support worker and participant to use together to plan the primary care appointment. We have described this in more detail in the description of the intervention.

Reviewer: how will the intervention help people without robust social networks?

Response: in two ways: 1) the PSW will make suggestions for ways to expand the person’s social network; 2) the observations, supervision notes and qualitative feedback will identify whether additions to the intervention are needed to try to build social networks.

Reviewer: Proposed data analysis – no evaluation which aims to detect differences before and after the intervention.

Response: the National Institute for Health Research (NIHR), who funded the study, advise against using feasibility trial effect sizes to calculate a sample size for an effectiveness trial. The confidence intervals are too wide.

Reviewer 3

Reviewer: target population unclear.

Response: we have added more information about the teams we recruited from under ‘settings’ and the patient group they care for.

Reviewer: how is the current manuscript different from other peer-support interventions?

Response: We have clarified in the paper that one major difference is that this pilot is occurring in the UK. The UK has very different health systems and peer support services to the US, where much of the research has been conducted. Further, the intervention being delivered by the PSWs and encouraging a primary care appointment is novel.

Reviewer: many outcome measures listed but feasibility appears main aim.

Response: see last point and response to Reviewer 2.

VERSION 2 – REVIEW

REVIEWER	Dr. Candelaria Mahlke, PI University Medical Center Hamburg-Eppendorf (UKE), Germany
REVIEW RETURNED	12-Dec-2018

GENERAL COMMENTS	thank you very much for the nice revision of the article and overall this interesting project! I think the manuscript is ready for publication and hope it will reach a lot of readers.
---

REVIEWER	Erin Kelly University of Southern California
REVIEW RETURNED	07-Nov-2018

GENERAL COMMENTS	Second round review of entitled “Managing transition from secondary mental to primary health care using peer-delivered well-being network mapping: protocol for a pilot study” The manuscript was much improved from the first round. Conceptually, I think the idea of building your intervention around the use of peers and social networks could be very useful for this population. My main concern for the design of your intervention is that 4 months might be too short of a period of time to effect significant change, as this is a population that can be slow to engage around their health and healthcare. A second concern is regarding what you have chosen to measure relative to your stated expectations of the mechanisms and outcomes you will influence. In Figure 1, the authors underlined the domains that they are not measuring (which was very helpful), but that highlighted some issues. I would encourage the researchers to either reconsider including an outcome in their figure that they don't intend to measure or think about including a measure of it. This particularly important given this statement “The first goal was to visit a primary care service for a physical health appointment; the second was to create a behaviour change with a view to improving physical health.” If the second goal is to change health behavior – you might consider including a measure of that outcome. Similarly, please also consider the implications of not measuring most of the mechanisms that you expect to underlie your model. I was also confused that you include a measure of self-efficacy that you don't think corresponds to any of the self-efficacy mechanisms that you have underlined. A few other comments: Per my first review, my first suggestion would be to adjust the title of the paper to something less convoluted. – Ex. Development of a peer-led, network-mapping intervention to improve the health of individuals with severe mental illnesses The definition of secondary care mental health services is not a universal term. Please define for an international audience (could be put in parentheses after the first use of the term). Adding in subheadings into your introduction could help. Thank you for providing an example of the social network. Can you add a legend for the colors of the dots as well? Methods section – what kinds of demographics will you collect and what kinds of diagnostic info?
--

	p.11 - “possess a high risk of harm to others” is this supposed to be “pose a high risk of harm to others”? Page 15 – feasibility of the intervention should also include your second goal of completion of their health goal – not just going to the doctor. A few grammatical suggestions - There are numerous singular-plural disagreements throughout the paper (see page 5 – “These functions may be carried out by professionals is specialist mental health care, however caseload sizes may not permit the time needed to focus on physical health and those whose mental health is fairly stable are likely to be discharged to primary care (19).”) and several odd choices of prepositions (For example, p4, line 46 “Attitudes to people”). Community-based instead of community based Peer-led instead of peer led
--	--

VERSION 2 – AUTHOR RESPONSE

Reviewer The manuscript was much improved from the first round. Conceptually, I think the idea of building your intervention around the use of peers and social networks could be very useful for this population. My main concern for the design of your intervention is that 4 months might be too short of a period of time to effect significant change, as this is a population that can be slow to engage around their health and healthcare. A second concern is regarding what you have chosen to measure relative to your stated expectations of the mechanisms and outcomes you will influence. In Figure 1, the authors underlined the domains that they are not measuring (which was very helpful), but that highlighted some issues. I would encourage the researchers to either reconsider including an outcome in their figure that they don't intend to measure or think about including a measure of it. This particularly important given this statement “The first goal was to visit a primary care service for a physical health appointment; the second was to create a behaviour change with a view to improving physical health.” If the second goal is to change health behavior – you might consider including a measure of that outcome.

Similarly, please also consider the implications of not measuring most of the mechanisms that you expect to underlie your model.

I was also confused that you include a measure of self-efficacy that you don't think corresponds to any of the self-efficacy mechanisms that you have underlined.

Response:

At this stage we are not in a position to change our outcome measures. Besides this consideration, to add measures would place undue burden on participants, particularly as the sample size does not allow for hypothesis testing. This phase of the study allows for qualitative data collection, including regarding behavior changes, which will guide the choice of measures for the next phase (a larger study) through refinement of the model of the intervention.

A few other comments:

Per my first review, my first suggestion would be to adjust the title of the paper to something less convoluted. – Ex. Development of a peer-led, network-mapping intervention to improve the health of individuals with severe mental illnesses

Response: Thank you for this suggestion. We have amended accordingly

Reviewer: The definition of secondary care mental health services is not a universal term. Please define for an international audience (could be put in parentheses after the first use of the term).

Response: We have used community mental health teams or specialist mental health care depending on the context.

Reviewer: Adding in subheadings into your introduction could help.

Response: We have added some suggested headings for the journal to consider.

Reviewer Thank you for providing an example of the social network. Can you add a legend for the colors of the dots as well?

Response: we have done this

Reviewer: Methods section – what kinds of demographics will you collect and what kinds of diagnostic info?

Response: we have added the demographic information to be collected. As clinical diagnosis is an eligibility criterion we have added this to the inclusion criteria.

Reviewer: “possess a high risk of harm to others” is this supposed to be “pose a high risk of harm to others”? -

Response: We have corrected this

Reviewer: Page 15 – feasibility of the intervention should also include your second goal of completion of their health goal – not just going to the doctor.

Response: This is assessed qualitatively (see response to first comment).

Reviewer:

A few grammatical suggestions - There are numerous singular-plural disagreements throughout the paper (see page 5 – “These functions may be carried out by professionals is specialist

mental health care, however caseload sizes may not permit the time needed to focus on physical health and those whose mental health is fairly stable are likely to be discharged to primary care (19).”)

and several odd choices of prepositions (For example, p4, line 46 “Attitudes to people”).

Community-based instead of community based

Response: these have been corrected